# Cross-Lingual Named Entity Recognition via Wikification

## Abstract

Named Entity Recognition (NER) models for language $L$ are typically trained using annotated data in that language. We study *cross-lingual NER*, where a model for NER in $L$ is trained on a source language (or multiple source languages). We introduce a language independent method for NER, building on cross-lingual wikification, a technique that grounds words and phrases in a non-English text into English Wikipedia entries. Thus, mentions in text in any language can be described using a set of categories and FreeBase types, yielding, as we show, strong language-independent features. With this insight, we propose an NER model that can be applied to all languages in Wikipedia. When trained on English, our model outperforms comparable approaches on the standard CoNLL datasets (Spanish, German, and Dutch) and also performs very well on low-resource languages (Turkish, Tagalog, Yoruba, Bengali, and Tamil) that have significantly smaller Wikipedia. Moreover, our methods allows us to train on multiple source languages, typically improving NER results on the target languages. Finally, we show that our language-independent features can be used also to enhance monolingual NER systems, yielding improved results for all 9 languages.

## 1 Introduction

Named Entity Recognition (NER) is the task of identifying and typing phrases that contain the names of persons, organizations, locations, and so on. It is an information extraction task that is important for understanding large bodies of text and is considered an essential pre-processing stage in Natural Language Processing (NLP) and Information Retrieval systems.

NER is successful for languages which have a large amount of annotated data, but for languages with little to no annotated data, this task becomes very challenging. There are two common approaches to address the lack of training data problem. The first approach is to automatically generate annotated training data in the target language from Wikipedia articles or from parallel corpora. The performance of this method depends on the quality of the generated data and how well the language-specific features are explored. The second approach is to train a model on another language which has abundant training data, and then apply the model directly on test documents in the target language. This direct transfer technique relies on developing language-independent features. Note that these two approaches are orthogonal and can be used together.

In this paper, we focus on the direct transfer setting. We propose a cross-lingual NER model which is trained on annotated documents in one or multiple source languages, and can be applied to all languages in Wikipedia. The model depends on a cross-lingual wikifier, which only requires multilingual Wikipedia, no sentence-aligned or word-aligned parallel text is needed.

Recently, much attention has been given to cross-lingual wikification and entity linking research (Ji et al., 2015; Ji et al., 2016; Moro et al., 2014; Tsai and Roth, 2016). The key contribution of the current paper is the development of a method that makes use of cross-lingual wikification to generate language-independent features for NER, and showing how useful this can be to training NER models with no annotation in the target language. Given a mention (sub-string) from a document written in a foreign language, the goal

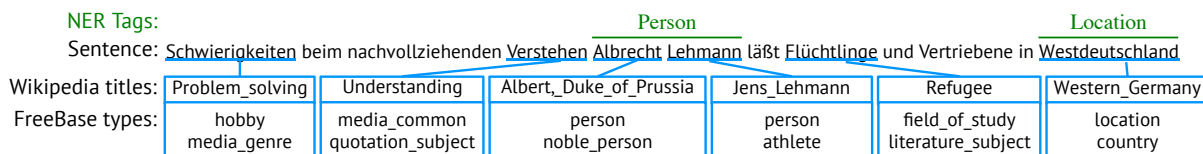

Figure 1: An example of German sentence. We ground each word to the English Wikipedia by a cross-lingual wikifier. A word is not linked if it is a stop word or the wikifier returns NIL. We can see that the FreeBase types are strong signals to NER even with imperfect disambiguation.

of cross-lingual wikification is to find the corresponding title in the English Wikipedia. Traditionally, wikification has been considered a downstream task of NER. That is, a named entity recognizer is firstly applied to identify the mentions of interest, and then the wikifier grounds the extracted mentions to Wikipedia entries. In contrast to this traditional pipeline, we show that the ability to disambiguate words is very useful in identifying named entities. By grounding every $n$-gram to the English Wikipedia, we can obtain useful clues regardless of the language used in testing documents.

Figure 1 shows an example of a German sentence. We use a cross-lingual wikifier to ground each word to the English Wikipedia. We can see that even though disambiguation is not perfect, the FreeBase types still provide valuable information. That is, although "Albrecht Lehmann" is not an entry in Wikipedia, the wikifier still links "Albrecht" and "Lehmann" to people. Since words in any language are grounded to the English Wikipedia, the corresponding Wikipedia categories and Freebase types can be used as language-independent features.

The proposed model significantly outperforms comparable direct transfer methods on Spanish, Dutch, and German CoNLL data. We also evaluate the model on five low-resource languages: Turkish, Tagalog, Yoruba, Bengali, and Tamil. Due to small sizes of Wikipedia, the overall performance is not as good as the CoNLL experiments. Nevertheless, the wikifier features still give significant improvement, and the proposed direct transfer model outperforms the state of the art, which assumes parallel text and some interaction with a native speaker of the target language. In addition, we show that the proposed language-independent features not only perform well on the direct transfer scenario, but also improve monolingual models, which are trained on the target language. Another advantage of the proposed direct transfer model is that we can train on documents from multiple languages together, and further improve the results.

## 2 Related Work

There are three main branches of work for extending NLP systems to many languages: projection across parallel data, Wikipedia-based approaches, and direct transfer. Projection and direct transfer take advantage of the success of NLP tools on high-resource languages. Wikipedia-based approaches exploit the fact that by editing Wikipedia, thousands of people have made annotations in hundreds of languages.

### 2.1 Projection

Projection methods take a parallel corpus between source and target languages, annotate the source side, and push annotations across learned alignment edges. Assuming that source side annotations are high quality, success depends largely on the quality of the alignments, which depends, in turn, on the size of the parallel data, and the difficulty of aligning with the target language.

There is work on projection for POS tagging (Yarowsky et al., 2001; Das and Petrov, 2011; Duong et al., 2014), NER (Wang and Manning, 2014; Kim et al., 2012; Ehrmann et al., 2011), and parsing (Hwa et al., 2005; McDonald et al., 2011).

Wang and Manning (2014) show that projecting expectations of labels instead of hard labels can improve results. They experiment in two different settings: weakly-supervised, where only parallel data is available, and semi-supervised, where annotated training data is available along with unlabeled parallel data.

### 2.2 Using Wikipedia

Wikipedia has been used for a large number of NLP tasks, from use as a semantic space (Gabrilovich and Markovitch, 2007; Song and Roth, 2014), to generating parallel data (Smith et

al., 2010), to use in open information extraction (Wu and Weld, 2010). It has also been used to extract training data for NER, under the intuition that Wikipedia is already (partially) annotated with NER labels, in the form of links to pages. Nothman et al. (2012) generate *silver-standard* NER data from Wikipedia using link targets, and other heuristics. This can be gathered for any language in Wikipedia, but several of the heuristics depend on language-specific rules. Al-Rfou et al. (2015) generate training data from Wikipedia articles using a similar manner. The polyglot word embeddings (Al-Rfou et al., 2013) are used as features in their NER model. Although the features are delexicalized, the embeddings are unique to each language, and so the model cannot transfer.

Kim et al. (2012) use Wikipedia to generate parallel sentences with NE annotations. They propose a semi-CRF model for aligning entities in parallel sentences. Results are very strong on Wikipedia data. This is a hybrid approach in that it is supervised projection using Wikipedia.

Our work is most closely related to Kazama and Torisawa (2007). They do NER using Wikipedia category features for each mention. However, their method for wikifying text is not robust to ambiguity, and they only do monolingual NER.

Sil and Yates (2013) create a joint model for NER and entity linking (EL) in English. They avoid the traditional pipeline of NER then EL by overgenerating mentions in the first stage and using NER features to rank candidates. While the results are promising, the model is not scalable to other languages because it requires a trained NER system and NP chunker.

### 2.3 Direct Transfer

The idea of direct transfer is to train a model in a high-resource setting using delexicalized features, that is, features that do not depend on word forms, and to directly apply it to text in a new language.

Täckström et al. (2012) experiments with direct transfer of dependency parsing and NER, and shows that using word cluster features can help, especially if the clusters are forced to conform across languages. The cross-lingual word clusters are induced using large parallel corpora.

Building on this work, Täckström (2012) focuses solely on NER, and includes experiments on self-training, and multi-source transfer for NER. Their experiments are orthogonal to ours, and

| Base features | | |
|---|---|---|
| *Non-Lexical* | | |
| Previous Tags | $(t_{i-1}, t_{i-2})$ | |
| Tag Context | (distr. for $[w_i, w_{i+1}, w_{i+2}]$) | |
| *Lexical* | | |
| Forms | $(..., w_{i-1}, w_i, w_{i+1}, ...)$ | |
| Affixes | (prefixes and suffixes of $w_i$) | |
| Capitalization | ($w_i$ capitalized?) | |
| Prev. Tag Pattern | $(t_{i-2}, w_{i-1}, w_i)$ | |
| Word type | (capital? digits? letter?) | |
| **Gazetteers** | | |
| Multilingual Wikipedia titles | | |
| **Cross-lingual Wikifier Features** | | |
| Freebase types of $(w_{i-1}, w_i, w_{i+1})$ | | |
| Wikipedia categories of $(w_{i-1}, w_i, w_{i+1})$ | | |

Table 1: Feature groups. Base features are the features used by Ratinov and Roth (2009), the state of the art English NER model. Gazetteers and cross-lingual wikifier features are described in detail in Section 3.

could be combined nicely. This work is closest to ours in terms of method, and so we compare against it in our experiments.

Our work falls under the umbrella of direct transfer methods combined with use of Wikipedia. We introduce wikifier features, which are truly delexicalized, and use Wikipedia as a source of information for each language.

## 3 Named Entity Recognition Model

We use the state of the art English NER model from Ratinov and Roth (2009) as a the base model. This model approaches NER as a multiclass classification problem with greedy decoding, using the BIO labeling scheme. The underlying classifier is averaged perceptron.

Table 1 summarizes the features used in our model. These can be divided into a base set of standard features which are included in Ratinov and Roth (2009), a set of gazetteer features which are based on titles in multilingual Wikipedia, and our novel cross-lingual wikifier features. The base set of features can be further divided into non-lexical and lexical categories.

### 3.1 Base Features

**Non-Lexical Features** Ratinov and Roth (2009) uses a small number of non-lexical features. For example, the previous tag feature is useful in pre-

dicting I- tags, because the previous tag should never be an O. The tag context feature looks in a 1000 word history and gathers statistics over tags assigned to words $[w_i, w_{i+1}, w_{i+2}]$. These features are included in all experiments.

In contrast with (Täckström et al., 2012), we do not use POS tags as features. We could not get the universal POS tags for all languages in our experiments, and an earlier experiment indicated that adding POS tags do not improve the performance due to the accuracy of tagger.

**Lexical Features** Lexical features are very important for monolingual NER. In the direct transfer setting, lexical features are useful if the target language is close to the training language. We use a small number of simple features, including word forms, affixes, capitalization, and tag patterns. The latter feature looks at a small window of text (at most 2 tokens) before the word in question. If there is a named entity in the window, it makes a feature out of NETag$+w_{i-2} + w_{i-1}$. Word type features simply indicate whether the word in question is all capitalized, is all digits, or is all letters.

### 3.2 Gazetteer Features

One of the larger performance improvements in Ratinov and Roth (2009) came from the use of gazetteers. We include gazetteers also in our model, except we gather them in each language from Wikipedia. As in Ratinov and Roth (2009), we use the gazetteers as features in the model. Specifically, we group gazetteers by topic, and use the name of the gazetteer file as the feature.

The method is to iteratively extend a short window to the right of the word in question. As the window increases in size, we search all gazetteers for occurrences of the phrase in the window. If we find a match, we add a feature to each word in the phrase according to its position in the phrase, either B for beginning, I for inside, or L for last. If the phrase is a single word, it is given a U feature.

This method generalizes gazetteers to unseen entities. For example, given the phrase "Bill and Melinda Gates Foundation", "Bill" is marked as both B-PersonNames and B-Organizations, while "Foundation" is marked as L-Organizations. Imagine encountering at test time a fictional organization called "Dave and Sue Harris Foundation." Although there is no gazetteer that contains this name, we have learned that "B-PersonName and B-PersonName B-PersonName Foundation" is a

strong signal for an organization.

### 3.3 Cross-lingual Wikifier Features

As shown in Figure 1, disambiguating words to Wikipedia entries allows us to obtain useful information for NER from the corresponding FreeBase types and Wikipedia categories. A cross-lingual wikifier grounds words and phrases of non-English languages to the English Wikipedia, which provides language-independent features for transferring an NER model directly.

We use the system proposed in Tsai and Roth (2016), which grounds input strings to the intersection of the English Wikipedia and the target language Wikipedia. The only requirement of this system is the multilingual Wikipedia dump and it can be applied on all languages in Wikipedia.

Since we want to ground every $n$-gram ($n \leq 4$) in the document, deviating from the normal usage that only considers few mentions of interest, we modify the system in the following two ways:

- The original candidate generation process queries the index by both whole input string and the individual tokens of the string. For the $n$-grams where $n > 1$, we generate title candidates only according to the whole string, not individual tokens. For instance, if it is allowed to generate title candidates based on individual tokens, the bigram "in Germany" will be linked to the title Germany thus wrongly considered as a named entity.

- The original ranking model includes the embeddings of other mentions in the document as features. It is clear that if we know what other important entities exist in the document, they provide useful clues to disambiguate a mention. However, if we want to wikify all $n$-grams, it makes no sense to include all of them as features, since the ranking model has already included features from TF-IDF weighted context words.

After wikifying every $n$-gram [1], we set the types of each $n$-gram as the coarse- and fine-grained FreeBase types and Wikipedia categories from the top 2 title candidates returned by wikifier. For each word $w_i$, we use the types of $w_i$, $w_{i+1}$, and $w_{i-1}$, and the types of the $n$-grams which contain $w_i$ as features. Moreover, we also include the

---

[1] We set $n$ to 4 in all our experiments.

ranker features in wikifier from the top candidate as features. This could serve as a linker (Ratinov et al., 2011), which rejects the top prediction if it has a low confidence.

# 4 Experiments and Analysis

In this section, we conduct experiments to validate and analyze the proposed NER model. First, we show that adding wikifier features improves results on monolingual NER. Second, we show that wikifier features are strong signals in direct transfer of a trained NER model across languages. Finally, we explore the importance of Wikipedia size to the quality of wikifier features and study using multiple source languages.

## 4.1 Datasets

We use data from CoNLL2002/2003 shared tasks (Tjong Kim Sang, 2002; Tjong Kim Sang and De Meulder, 2003). The 4 languages represented are English, German, Spanish, and Dutch, each annotated using the IOB1 labeling scheme, which we convert to the BIO labeling scheme. All training is on the train set, and testing is on the test set. The evaluation metric for all experiments is phrase level F1, as explained in (Tjong Kim Sang, 2002).

In order to experiment on a broader range of languages, we also use data from the REFLEX (Simpson et al., 2008) and LORELEI projects. From LORELEI, we use Turkish,[2] From RE-FLEX, we use Bengali, Tagalog, Tamil, and Yoruba.[3] While Turkish, Tagalog, and Yoruba each have a few non-Latin characters, Bengali and Tamil are with an entirely non-Latin script. This is a major reason for inclusion in our experiments. We use the same set of test documents as used in Zhang et al. (2016). All other documents in the REFLEX and LORELEI packages are used as the training documents in our monolingual experiments. We refer to these five languages collectively as low-resource languages.

Besides PER, LOC, and ORG, some low-resource languages contain TIME tags and TTL tags, which represented titles in text, such as Secretary, President, or Minister. Since such words are not tagged in CoNLL training data, we opted to simply remove these tags. On the other hand, there is no MISC tag in the low-resource languages. Instead, many MISC-tagged entities in the

CoNLL datasets have LOC tags in the REFLEX and LORELEI packages, e.g., Italian and Chinese. We modify a MISC-tagged word to LOC tag if it is grounded to an entity with location as a FreeBase type, and remove all the other MISC tags in the training data. This process of changing MISC tags is only done when we train on CoNLL documents and test on low-resource languages.

The only requirement to build the cross-lingual wikifier model is a multilingual Wikipedia dump, and it can be trivially applied to all languages in Wikipedia. The top section of Table 2 lists Wikipedia sizes in terms of articles,[4] the number of titles linked to English titles, and the number of training and test mentions for each language.

Besides the English gazetteers used in Ratinov and Roth (2009), we collect gazetteers for each language using Wikipedia titles. A Wikipedia title is included in the list for person names if it contains FreeBase type person. Similarly, we also create a location list and an organization list for each language. The total number of names in the gazetteers of each language is listed in Table 2.

## 4.2 Monolingual Experiments

We begin by showing that wikifier features help when we train and test on the same language. The middle section of Table 2 shows the results.

In the 'Wikifier only' row, we use only wikifier features and previous tags features. This is intended to show the predictive power of wikifier features alone. Without using any lexical features, it gets good scores on the languages that have a large Wikipedia. These numbers represent the quality of the cross-lingual wikifier in that language, which in turn is correlated with the size of Wikipedia and size of the intersection with English Wikipedia.

The next row, 'Base features', shows that lexical features are always better than wikifier features only. This squares with the common wisdom that lexical features are important for NER.

Adding gazetteers to the base features improves more than 3 points for every language except Bengali and Tamil. Since these two languages use entirely non-Latin scripts, the words will not match any names in the English gazetteers, which have higher coverage than other languages' gazetteers.

Finally, the '+Wikifier' row shows that our pro-

---

[2]LDC2014E115
[3]LDC2015E13,LDC2015E90,LDC2015E83,LDC2015E91

[4]From `https://en.wikipedia.org/wiki/List_of_Wikipedias`, retrieved March 2016

| APPROACH | Latin Script | | | | | | | Non-Latin Script | | |
| --- | --- | --- | --- | --- | --- | --- | --- | --- | --- | --- |
| | EN | NL | DE | ES | TR | TL | YO | BN | TA | AVG |
| Wiki size | 5.1M | 1.9M | 1.9M | 1.3M | 269K | 64K | 31K | 42K | 85K | - |
| En. intersection | - | 755K | 964K | 757K | 169K | 49K | 30K | 34K | 51K | - |
| Gazetteer size | 8.5M | 579K | 1M | 943K | 168K | 54K | 20K | 29K | 10K | - |
| Entities (train) | 23.5K | 18.8K | 11.9K | 13.3K | 5.1K | 4.6K | 4.1K | 8.8K | 7.0K | - |
| Entities (test) | 5.6K | 3.6K | 3.7K | 3.9K | 2.2K | 3.4K | 3.4K | 3.5K | 4.6K | - |
| Monolingual Experiments | | | | | | | | | | |
| Wikifier only | 72.90 | 57.38 | 51.95 | 59.33 | 52.63 | 51.41 | 33.63 | 45.96 | 37.83 | 51.45 |
| Base Features | 85.24 | 77.34 | 65.11 | 79.98 | 65.21 | 74.34 | 54.78 | 69.11 | 55.30 | 69.60 |
| +Gazetteers | 88.92 | 82.69 | 68.74 | 83.51 | 70.67 | 77.53 | 57.89 | 69.50 | 56.76 | 72.91 |
| +Wikifier | **89.47** | **84.90** | **72.97** | **84.25** | **73.50** | **78.18** | **59.27** | **70.62** | **60.00** | **74.80** |
| Direct Transfer Experiments | | | | | | | | | | |
| Wikifier only | | 36.66 | 38.55 | 40.04 | 43.09 | 36.97 | 25.09 | **41.81** | **27.85** | 36.26 |
| Base Features | | 44.10 | 25.24 | 41.81 | 30.50 | 50.45 | 32.48 | 2.30 | 1.74 | 28.58 |
| +Gazetteers | | 49.66 | 35.06 | 55.04 | 30.90 | 64.07 | 34.42 | 3.14 | 0.30 | 34.07 |
| +Wikifier | | **62.10** | **47.14** | **60.97** | **48.41** | **65.32** | **36.79** | 6.72 | 2.99 | **41.31** |
| Täckström baseline | | 48.4 | 23.5 | 45.6 | - | - | - | - | - | - |
| Täckström bitext clusters | | 58.4 | 40.4 | 59.3 | - | - | - | - | - | - |
| Zhang et al. (2016) | | - | - | - | 43.6 | 51.3 | 36.0 | 34.8 | 26.0 | 38.3 |

Table 2: **Data sizes, monolingual experiments, and direct transfer experiments**. Wiki size is the number of articles in Wikipedia. For monolingual experiments, we train the proposed model on the training data of the target languages. 'Wikifier only' uses the previous tags features also. For direct transfer experiments, all models are trained on CoNLL English training set. The rows marked Täckström come from (Täckström et al., 2012), and are the baseline and clustering result. The plus signs (+) signify cumulative addition. EN: English, NL: Dutch, DE: German, ES: Spanish, TR: Turkish, TL: Tagalog, YO: Yoruba, BN: Bengali, TA: Tamil.

posed features are valuable even in combination with strong features. It improves upon base features and gazetteer features for all 9 languages. These numbers may be less than state of the art because the features we use are designed for English, and may not capture lexical subtleties in every language. Nevertheless, they show that wikifier features have a non-trivial signal that has not been captured by other features.

### 4.3 Direct Transfer Experiments

We evaluate our direct transfer experiments by training on English and testing on the target language. The results from these experiments are shown in the bottom section of Table 2.

The 'Wikifier only' row shows that the wikifier features alone preserve a signal across languages. Interestingly, for both Bengali and Tamil, this is the strongest signal, and gets the highest score. If the lexical features are included when we train the English model, the learning algorithm will give them too much emphasis, thus decreasing the importance of the wikifier features. Since Bengali and Tamil use non-Latin scripts, no lexical feature in English will fire at test time. Thus, approaches that include base features perform poorly.

The results of 'Base features' can be viewed as a sort of language similarity to English, which, in this case, is related to lexical overlap and similarity between the scripts. Comparing to monolingual experiments, we can see that the lexical features become weak in the cross-lingual setting.

The gazetteer features are again shown to be very useful for almost all languages except Bengali and Tamil due to the reason explained in the monolingual experiment and to the inclusion of lexical features. For the rest of languages, the improvement from adding gazetteers is even more than the improvement in the monolingual setting.

For nearly every language, wikifier features help dramatically, which indicates that they are very good delexicalized features. It adds more than 10 points on Dutch, German, and Turkish.

The trend in Table 2 suggests the following strategy when we want to extract named entities in a new foreign language: It is better to include all features if the foreign language uses Latin script, since the names are likely mentioned using the

same way as in English. Otherwise, using wiki-fier features could be the best setting.

Täckström et al. (2012) also directly transfer an English NER model using the same setting as ours: train on the CoNLL English training set and predict on the test set of other three languages. We compare our baseline transfer model (Base Features) to the row denoted by "Täckström baseline". Even though we do not use gold POS tags, we see that our results are comparable. The second Täckström row uses parallel text to induce multilingual word clustering. While this approach is orthogonal to ours, and could be used in tandem to get even better scores, we compare against it for lack of a more closely aligned scenario. We see that for each language, we significantly outperform their approach.

We note that our numbers are comparable to those reported for WIKI-2 in Nothman et al. (2012) for the CoNLL languages (with the exception of German, where theirs is higher). Their system requires language-specific heuristics to generate their *silver-standard* training data from Wikipedia articles. What they gain for single languages, they likely lose in generalizability. This approach is orthogonal to ours and we can also use their silver-standard data in training.

For the low-resource languages, we compare our direct transfer model with the expectation learning model proposed in Zhang et al. (2016). This model is not a direct transfer model, but it does not use any training data in the target languages either. Instead, for each target language, it generates patterns from parallel documents between English and the target language, a large monolingual corpus in the target language, and one-hour interaction with a native speaker of the target language. Note that they also use a cross-lingual wikifier, but only for refining the entity types. On the other hand, in our model, the features from the wikifier are used in both detecting entity mention boundaries and the entity types. We can see that our approach performs better than their model on all five languages even though we assume much fewer resources. The difference is most significant on Turkish, Tagalog, and Bengali.

### 4.4 Quality of Wikifier Features

One immediate question is why wikifier features are less helpful on the low-resource languages results than on the CoNLL languages? In this exper-

| FEATURES | SPANISH | | GERMAN | |
|---|---|---|---|---|
| | #inter. | F1 | #inter. | F1 |
| Wikifier only | 757K | 40.04 | 964K | 38.55 |
| W.−FB query | 757K | 34.69 | 964K | 28.27 |
| W.−FB−50% inter. | 379K | 30.32 | 482K | 27.24 |
| W.−FB−90% inter. | 76K | 29.44 | 96K | 25.94 |

Table 3: The F1 scores of using only wikifier features with removing the support from FreeBase and varying the number of titles linked to the English Wikipedia. 'W.−FB query' removes the component of querying FreeBase by the target language title from 'Wikifier only'. '−$X\%$ inter.' indicates removing $X\%$ of the interlanguage links with English titles. The column #inter. shows the number of titles that intersect with English.

iment, we show that smaller Wikipedia sizes result in worse Wikipedia features, which is the reason Yoruba has bad 'Wikifier only' results and the improvement from the wikifier features is much smaller.

The cross-lingual wikifier that we use in our system only grounds words to the intersection of the English and target language Wikipedia. Given a Wikipedia title in the target language, we first retrieve FreeBase ID by querying FreeBase API. If it fails, we find the corresponding English Wikipedia title via interlanguage links and then query the API with the English title. However, FreeBase does not contain entities in Yoruba, Bengali, and Tamil, so the first step will always fail for these three languages. We remove this step in the experiments of high-resource languages and the results are shown in the row 'W.−FB query ' of Table 3. We can see that the performance drops a lot, because many words have no features from FreeBase types.

Next, we randomly remove 50% and 90% of the interlanguage links to English titles. This will not only reduce the number of fired features from Wikipedia categories, but also FreeBase types since English titles are used to query FreeBase IDs. When 90% of interlanguage links are removed, the scores of Spanish and German are closer to Yoruba's score (27.85).

### 4.5 Training Languages

In all previous experiments, the training language is always English. In order to test the efficacy of training with languages other than English, we create a train/test matrix with all combinations of languages, as seen in Figure 2.

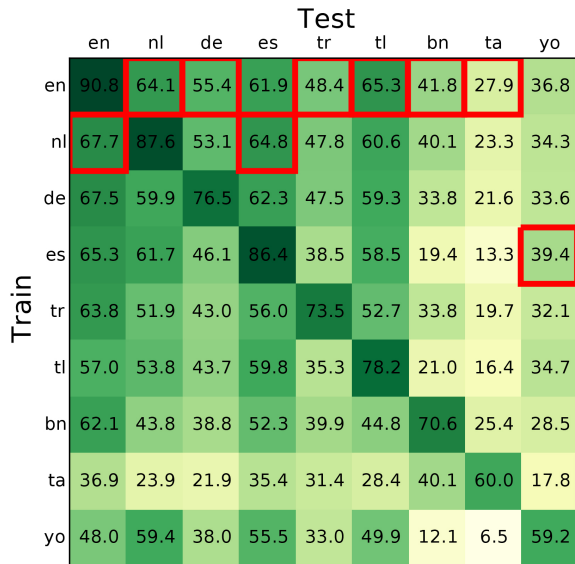

Figure 2: Different training/test language pairs. Scores shown are the F1 scores. The red boxes signify the best non-target training languages.

| TRAINING LANG | TR | TL | YO | AVG |
|---|---|---|---|---|
| EN | 48.41 | 65.32 | 36.79 | 50.17 |
| EN+ES | 45.33 | 67.06 | **38.75** | 50.38 |
| EN+NL | 49.74 | 65.84 | 37.38 | 50.99 |
| EN+DE | 48.49 | 65.32 | 36.79 | 50.20 |
| EN+ES+NL+DE | 49.02 | 66.94 | 37.73 | 51.23 |
| ALL−Test Lang | **50.17** | **67.32** | 37.31 | **51.60** |

Table 4: The F1 scores of the proposed direct transfer model on three low-resource languages using training data in multiple languages. The row "ALL−Test Lang" trains the model on all languages except the test language, Bengali, and Tamil. Bengali and Tamil are excluded since we use all features in this experiment.

The vertical axis represents training language, and the horizontal axis represents test language. A darker color signifies a higher score. For example, if we train on Spanish (es) and test on Yoruba (yo), we get an F1 of 39.4. When the test language is Bengali (bn) or Tamil (ta), we only use wikifier features. For other test languages, all features are included. Note that we ignore all MISC tags in the CoNLL languages (en, nl, de, es) in evaluation, since there is no MISC tag in the low-resource languages. The diagonals represent the monolingual setting in which we use all features for all languages. Since we are interested in transferring a model, we ignore the diagonals, and identify the best training language for a given test language as the largest off-diagonal in each column. These are demarcated with red boxes.

English is the best for most languages, with the exception of Dutch, best for Spanish, and Spanish, best for Yoruba. It makes sense that high-resource languages are better training languages because 1) there are more annotated training instances, 2) larger Wikipedia creates denser wikifier features, therefore providing better estimation of the weights to these features.

Table 4 shows the results of training on multiple languages. We use all features in this experiment. The row "EN" only trains the model on the English training documents, and the results are identical to those shown in Table 2. Adding

Spanish training data yields the best score on Yoruba, which agrees with Figure 2 where Spanish is the best training language for Yoruba. Using all CoNLL languages (EN+ES+NL+DE) adds more than 1 point F1 in average comparing to using English only. Finally, training on all but the test languages further improves the results.

This experiment shows that we can augment training data from other languages' annotated documents. Although the performance only increases a little, it does not hurt most of the time.

## 5 Conclusion and Discussion

We propose a language-independent model for cross-lingual NER using a cross-lingual wikifier to disambiguate every $n$-grams. This model works on all languages in Wikipedia and the only requirement is a Wikipedia dump. We study a wide range of languages in both monolingual and cross-lingual settings, and show significant improvements over strong baselines. An analysis shows that the quality of wikifier features depends on the Wikipedia size of the test language.

This work shows that if we can disambiguate words and phrases to the English Wikipedia, the typing information from Wikipedia categories and FreeBase are useful language-independent features for NER. However, there is other information from Wikipedia which we do not use, such as words from documents and relations between titles, which would require additional research.

In the future, we would like to experiment with including other techniques for multilingual NER that we discuss in Section 2 into our model, such as parallel projection and generating training data from Wikipedia automatically.

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
