# Peer review of "Cross-Lingual Named Entity Recognition via Wikification"

_CoNLL 2016 — decision unknown_

[Official Review · Reviewer 1 · rating 4 · confidence 4]
soundness 4 · originality 4 · clarity 5 · impact 3 · substance 4 · appropriateness 5 · meaningful comparison 4 · replicability 4 · presentation format Oral Presentation

This paper proposes an approach for multi-lingual named entity recognition
using features from Wikipedia. By relying on a cross-lingual Wikifier, it
identifies English Wikipedia articles for phrases in a target language and uses
features based on the wikipedia entry. Experiments show that this new feature
helps not only in the monolingual case, but also in the more interesting direct
transfer setting, where the English model is tested on a target language.

I liked this paper. It proposes a new feature for named entity recognition and
conducts a fairly thorough set of experiments to show the utility of the
feature. The analysis on low resource and the non-latin languages are
particularly interesting.

But what about named entities that are not on Wikipedia? In addition to the
results in the paper, it would be interesting to see results on how these
entities are affected by the proposed method. 

The proposed method is strongly dependent on the success of the cross-lingual
wikifier. With this additional step in the pipeline, how often do we get errors
in the prediction because of errors in the wikifier?

Given the poor performance of direct transfer on Tamil and Bengali when lexical
features are added, I wonder if it is possible to regularize the various
feature classes differently, so that the model does not become over-reliant on
the lexical features.

[Official Review · Reviewer 2 · rating 4 · confidence 4]
soundness 4 · originality 3 · clarity 4 · impact 3 · substance 4 · appropriateness 5 · meaningful comparison 4 · replicability 4 · presentation format Poster

This paper is concerned with cross-lingual direct transfer of NER models using
a very recent cross-lingual wikification model. In general, the key idea is not
highly innovative and creative, as it does not really propose any core new
technology. The contribution is mostly incremental, and marries the two
research paths: (1) direct transfer for downstream NLP tasks (such as NER,
parsing, or POS tagging), and (2) very recent developments in the cross-lingual
wikification technology. However, I pretty much liked the paper, as it is built
on a coherent and clear story with enough experiments and empirical evidence to
support its claims, with convincing results. I still have several comments
concerning the presentation of the work.

Related work: a more detailed description in related work on how this paper
relates to work of Kazama and Torisawa (2007) is needed. It is also required to
state a clear difference with other related NER system that in one way or
another relied on the encyclopaedic Wikipedia knowledge. The differences are
indeed given in the text, but they have to be further stressed to facilitate
reading and placing the work in context. 

Although the authors argue why they decided to leave out POS tags as features,
it would still be interesting to report experiments with POS tags features
similar to Tackstrom et al.: the reader might get an overview supported by
empirical evidence regarding the usefulness (or its lack) of such features for
different languages (i.e., for the languages for which universal POS are
available at least). 

Section 3.3 could contribute from a running example, as I am still not exactly
sure how the edited model from Tsai and Roth works now (i.e., the given
description is not entirely clear).

Since the authors mention several times that the approaches from Tackstrom et
al. (2012) and Nothman et al. (2012) are orthogonal to theirs and that they can
be combined with the proposed approach, it would be beneficial if they simply
reported some preliminary results on a selection of languages using the
combination of the models. It will add more flavour to the discussion. Along
the same line, although I do acknowledge that this is also orthogonal approach,
why not comparing with a strong projection baseline, again to put the results
into more even more context, and show the usefulness (or limitations) of
wikification-based approaches.

Why is Dutch the best training language for Spanish, and Spanish the best
language for Yoruba? Only a statistical coincidence or something more
interesting is going on there? A paragraph or two discussing these results in
more depth would be quite interesting.

Although the idea is sound, the results from Table 4 are not that convincing
with only small improvements detected (and not in all scenarios). A statistical
significance test reported for the results from Table 4 could help support the
claims.

Minor comments:

- Sect. 2.1: Projection can also be performed via methods that do not require
parallel data, which makes such models more widely applicable (even for
languages that do not have any parallel resources): e.g., see the work of
Peirsman and Pado (NAACL 2009) or Vulic and Moens (EMNLP 2013) which exploit
bilingual semantic spaces instead of direct alignment links to perform the
transfer.

- Several typos detected in the text, so the paper should gain quite a bit from
a more careful proofreading (e.g., first sentence of Section 3: "as a the base
model"; This sentence is not 'parsable', Page 3: "They avoid the traditional
pipeline of NER then EL by...", "to disambiguate every n-grams" on Page 8)